# Modeling Transposition of the Great Arteries with Patient-Specific Induced Pluripotent Stem Cells

**DOI:** 10.3390/ijms222413270

**Published:** 2021-12-09

**Authors:** Imelda Ontoria-Oviedo, Gabor Földes, Sandra Tejedor, Joaquín Panadero, Tomoya Kitani, Alejandro Vázquez, Joseph C. Wu, Sian E. Harding, Pilar Sepúlveda

**Affiliations:** 1Regenerative Medicine and Heart Transplantation Unit, Instituto de Investigación Sanitaria La Fe, 46026 Valencia, Spain; sandra.tejedorgascon@gmail.com (S.T.); avazsan@gmail.com (A.V.); 2National Heart and Lung Institute, Imperial College London, London W12 0NN, UK; g.foldes@imperial.ac.uk (G.F.); sian.harding@imperial.ac.uk (S.E.H.); 3Heart and Vascular Center, Semmelweis University, H1122 Budapest, Hungary; 4IGENOMIX S.L., Edificios Europark, Parque Tecnológico, 46980 Paterna, Spain; btcpar@gmail.com; 5Stanford Cardiovascular Institute, Stanford University School of Medicine, Stanford, CA 94305, USA; t-kitani@koto.kpu-m.ac.jp (T.K.); joewu@stanford.edu (J.C.W.)

**Keywords:** great arteries transposition, endothelial cells, angiogenesis, Notch signaling pathway, iPSC

## Abstract

The dextro-transposition of the great arteries (d-TGA) is one of the most common congenital heart diseases. To identify biological processes that could be related to the development of d-TGA, we established induced pluripotent stem cell (iPSC) lines from two patients with d-TGA and from two healthy subjects (as controls) and differentiated them into endothelial cells (iPSC-ECs). iPSC-EC transcriptome profiling and bioinformatics analysis revealed differences in the expression level of genes involved in circulatory system and animal organ development. iPSC-ECs from patients with d-TGA showed impaired ability to develop tubular structures in an in vitro capillary-like tube formation assay, and interactome studies revealed downregulation of biological processes related to Notch signaling, circulatory system development and angiogenesis, pointing to alterations in vascular structure development. Our study provides an iPSC-based cellular model to investigate the etiology of d-TGA.

## 1. Introduction

Congenital heart disease (CHD) represents almost one-third of all congenital diseases [1] and occurs in nearly 1% of live births [2]. The dextro-transposition of the great arteries (d-TGA) is one of the most common and severe types of CHD [1,3] with a prevalence of 20–33 per 100,000 newborns and an incidence of 5–8% of all CHD [4,5]. While d-TGA is most frequent in males (ratio 2:1), there is no correlation with any chromosomopathy [6].

d-TGA is generally reported with Heterotaxy syndrome, but the etiology and morphogenesis of TGA are still largely unknown. Nonetheless, several genes, such as *Pitx2* and *Tbx2*, have been evaluated in animal models [3,7]. During heart development, the cardiac tube expands and begins a morphogenetic looping process that ultimately brings the posterior part of the tube to a rostral position, dorsal to the outflow tract (OFT) [8]. During mid-development, the OFT is remodeled into separate pulmonary and aortic arteries, which involves direct interactions between myocardium, endocardium and neural crest cells [9]. Subendocardial extracellular matrix (ECM) provides physical support and signaling regulation to cells and regulates tissue morphology during embryonic development [10]. The correct rotation of the myocardial wall of the OFT is necessary for the normal position of the great arteries. A high proportion of Perlecan (*Hspg2*)-null mouse embryos present with complete TGA [11], suggesting the importance of the ECM and basement membranes in cardiac morphogenesis. d-TGA can also be induced by retinoic acid treatments [12], directly involving the *Tbx2-Tgfβ2* pathway [7].

Several studies have used animal models of TGA to better understand disease mechanisms, and there have only been a few studies reported in humans (reviewed in [3]). The emergence of human induced pluripotent stem cells (iPSC) [13] has raised the possibility of more robust disease modeling [14]. Due to the importance of endothelial cells (EC) in the vascular system, several studies have focused on designing robust protocols to differentiate iPSC into endothelial cells (iPSC-ECs) [15,16].

In the present study, we sought to generate a d-TGA cellular model based on the use of patient-specific iPSCs. Specifically, we generated two iPSC lines from patients with d-TGA, which were differentiated into iPSC-ECs. Several genes were identified by RNA-seq as potential players in the etiology of this congenital defect. We show that iPSC-ECs from patients with d-TGA have an impaired capacity to develop vascular networks concomitant with changes in the transcriptome, as assessed by bioinformatics analysis. Specifically, gene ontology (GO) biological processes related to embryonic morphogenesis/development, tube development and angiogenesis, together with Notch signaling pathways, were downregulated in EC derived from patients with d-TGA.

## 2. Results

### 2.1. Generation of Endothelial Cells from Differentiated iPSCs

We generated iPSC lines from two patients with d-TGA (TGA1 and TGA2) and also one control iPSC line from a healthy individual (Ctrl1), which were cultured under feeder-free conditions. We used the commercially available iPSC line IMR90-4 as a second control (Ctrl2). The cells were maintained undifferentiated and displayed typical morphology (Figure 1A) and normal karyotype (Appendix A). We differentiated the iPSC lines to ECs by the addition of different growth factors (Figure 1B), following a published protocol [15]. We then isolated cells that were double-positive for *CD31* and neuropilin-1 (*NRP-1*), a coreceptor of vascular endothelial growth factor (*VEGF*), at day 12, using fluorescence-activated cell sorting (FACS), which ensured an EC population. The results of the sorting analysis revealed a similar but low efficiency of EC differentiation in all four iPSC lines: FACS efficiency was 0.93 ± 0.69 (Ctrl1), 2.8 ± 2.08 (Ctrl2), 0.57 ± 0.23 (TGA1) and 0.6 ± 0.4 (TGA2). Isolated ECs showed a characteristic cobblestone morphology by brightfield microscopy (Figure 1C) and were able to grow up to four passages. During endothelial differentiation of iPSC, we obtained RNA samples from each cell line at days 0, 5, 12 and 19 for gene characterization by qPCR. As expected, the levels of the pluripotency gene *OCT4* decreased over time in all the lines tested. By contrast, the expression of the specific endothelial gene *CD31*, vascular-endothelial cadherin (*CDH5*) and receptor tyrosine kinase (*TIE2*) increased over time (Figure 1D). All iPSC lines exhibited an increase in *CD31* and *TIE2* expression over time, indicating the differentiation of iPSC into EC. The expression profile of *OCT4* and *CD31* was similar in all the iPSC lines; however, some differences were observed when *CDH5* and *TIE2* were analyzed (Figure 1C). 

### 2.2. RNA-Seq Analysis of Endothelial Cells from Differentiated Control and d-TGA iPSC Lines

We next performed RNA-seq and bioinformatics analysis of sorted double-positive cells at day 20 of differentiation of the four iPSC-EC lines. Despite the known variability that exists between individuals [17,18] and the different origin of the iPSC lines used, the iPSC-EC samples from controls and patients with d-TGA were well separated by the principal component analysis (Figure 2A). These results support the notion that despite the interindividual genotypic differences, there were robust genetic differences between control and TGA lines. 

A total of 4672 genes were identified by bioinformatics analysis, and 4471 genes had similar expression values in d-TGA and control groups. Identified genes with fewer than 150 counts were removed. We generated a volcano plot to visualize differentially expressed genes (DEG) (*p*-value ≤ 0.05 and fold change ≥ 2.0) in iPSC-ECs from d-TGA and control lines (Figure 2B). A total of 201 DEG were identified, of which 62 were upregulated in d-TGA lines and 139 were downregulated. Regarding the upregulated genes, we identified several genes associated with the extracellular matrix (ECM), such as collagen type I alpha 1 (*COL1A1*), *COL1A2* and laminin subunit alpha 4 (*LAMA4*). We also found an increase in the expression of the transcription factors *SIX1* and *TBX2*, which have important roles in heart morphogenesis [8,19]. A member of the Fox transcription factor family (*FOXC1*) with an important role in vascular development was also upregulated in iPSC-ECs from patients with d-TGA. 

Of the 139 downregulated genes in the d-TGA group, we identified several genes involved in vascular compartment and developmental processes. For example, specific endothelial markers, including *PECAM1*, *CDH5*, *TIE1* and VEGF receptor 2 (*KDR*) [20] were all downregulated in iPSC-ECs from patients with d-TGA, as well as genes involved in Notch signaling, including jagged canonical Notch ligand 2 (*JAG2*) and Delta-like canonical Notch ligand 4 (*DLL4*). In addition, genes involved in gap junctions, such as gap junction protein alpha 4 (*GJA4*) and *GJA5*, which are necessary to establish cell communication, were also downregulated. Likewise, vascular stabilizing genes related to the ECM, such as collagen type IV alpha 1 chain (*COL4A1*), *COL4A2* and *HSPG2*, were downregulated. A complete list of the upregulated and downregulated DEG is shown in Appendix A, respectively.

### 2.3. iPSC-ECs from Patients with d-TGA Show Alterations in Signaling Pathways

In agreement with the gene expression profiling, GO analysis of DEG demonstrated different biological processes associated with control and d-TGA iPSC-ECs. A list of all significantly overrepresented GO biological processes of upregulated and downregulated genes, as a function of their *p*-value, is shown in Appendix A, respectively. Many of the GO processes identified in upregulated genes were involved with developmental processes, including animal organ development and morphogenesis, embryonic development and morphogenesis, negative regulation of developmental process and anatomical structure development (Table 1). 

As a complementary representation, we used a REVIGO treemap to reduce the number of GO terms into ‘clusters’ and ‘superclusters’ of related terms, as described [21], where GO terms are visualized with different colors in rectangles representing the *p*-value (Figure 2C). The 304 GO biological processes identified were mainly grouped into two superclusters: ‘negative regulation of multicellular organismal process’, which was the supercluster most represented, and ‘animal organ development’. We performed the same enrichment analysis on the downregulated genes; 279 GO terms were identified (*p*-value < 0.05) and the most representative are listed in Table 2. Of note, many processes were related to blood vessel development and morphogenesis, tube development and morphogenesis, negative regulation of developmental process and regulation of blood vessel endothelial cell proliferation involved in sprouting angiogenesis.

These results suggest that the downregulated genes identified in iPSC-ECs from patients with d-TGA are of particular relevance for understanding the mechanisms of d-TGA in our system.

We also created a treemap to group the GO biological processes, which were mostly grouped into the ‘regulation of multicellular organismal process’ and ‘circulatory system development’ superclusters (Figure 2D).

Several GO biological processes were identified when the analysis was performed with up- and downregulated genes. GO analysis of genes downregulated in iPSC-ECs from patients with d-TGA *versus* controls revealed significant enrichment of genes involved in the circulatory system.

We next conducted an interactome analysis to study the interaction network between the targets of DEG in iPSC-ECs from patients with d-TGA *versus* controls (Figure 3). GO biological processes were identified with gene-set enrichment analysis, and the interaction network was visualized with Cytoscape software v3.7.2.

When targets of upregulated genes were analyzed, biological processes related to anatomical structure morphogenesis and development, regulation of the developmental process, embryonic morphogenesis and development were enriched in iPSC-ECs of patients with d-TGA (Table 3). 

When targets of downregulated genes were analyzed, biological processes related to cellular and animal organ development processes were enriched in iPSC-ECs of patients with d-TGA (Table 4). In agreement with the aforementioned results, processes related to circulatory system development, blood vessel development and morphogenesis, and Notch signaling were also identified.

### 2.4. Capillary-like Tube Formation Processes Are Altered in iPSC-ECs from Patients with d-TGA

The results of the bioinformatics analysis suggested that iPSC-ECs from patients with d-TGA might show defective blood vessel development and morphogenesis. To test this, we performed a functional in vitro assay with iPSC-ECs to evaluate their capacity to form a capillary-like tube network, as a readout of angiogenesis. Cells were seeded onto Matrigel in EGM-2 medium, and tube formation was evaluated microscopically after 3 h. Results showed that iPSC-ECs from patients with d-TGA were unable to correctly form tubular structures measured as total loops, total tube length, number of total tubes and branching points (Figure 4A,B), with significant differences in other parameters, including covered area (%), mean loop area and mean loop perimeter (Figure 4B). By contrast, the total net abundance was significantly higher in iPSC-ECs from patients with d-TGA, and no differences were found when the mean tube length was measured. These results are in line with the interactome analysis, where biological processes related to the regulation of angiogenesis were downregulated (Figure 3).

## 3. Discussion

d-TGA is the sixth most prevalent CHD from a total of 27 anatomical subtypes, as reported in a systematic review [22]. The advent of human iPSC technologies in 2007 by Yamanaka et al. [13] has been a major breakthrough for disease modeling and for therapeutic purposes [14], and detailed gene analysis of iPSC lines from patients with CHD is a promising strategy to study the mechanisms involved in the onset of cardiac diseases [23].

We studied the gene expression patterns in iPSC-ECs from patients with d-TGA, the deregulation of which may be related to the onset of disease. Two iPSC lines were successfully generated from patients with d-TGA and were differentiated into ECs. Despite the low efficiency of EC differentiation, the FACS-sorted *CD31+/NRP-1+* double-positive cells proliferated and expressed endothelial markers. We used bioinformatics analysis to identify genes differentially expressed in iPSC-ECs from patients with d-TGA. One of these genes was the transcription factor *Tbx2*, which was overexpressed in ECs from patients with d-TGA. *Tbx2* expression is closely related to heart development, as it is expressed in the heart tube, and it has been described to arrest cardiac development at looping [8]. It has also been observed to be highly expressed in the early stages of development, but it decreases over time [24]. Indeed, *Tbx2* expression is involved in other aspects of heart looping but is not related to chamber growth [25]. Another gene upregulated in iPSC-ECs from patients with d-TGA was *FOXC1*, which has been described to promote angiogenic activity in vascular endothelial cells that is determined by the correct balance between pro-angiogenic *FOXC1* activity in the endothelium and anti-angiogenic *FOXC1* activity in the surrounding cells [26]. Two members of the histone family, *HIST1H3C* and *HIST1H2BB*, were also upregulated in iPSC-ECs from d-TGA patients. Altered expression of different histones has been related to vascular diseases, such as abdominal aortic aneurysm, which is characterized by the degradation of the elastic media and remodeling of the aortic ECM [27]. Different genes upregulated in iPSC-ECs from d-TGA (*LAMA4*, *COL1A1*, *COL1A2* and *COL6A1*) are involved in ECM organization (GO:0030198). These data suggest that the expression of different genes related to ECM is altered in iPSC-ECs from patients with d-TGA. *HSPG2*, a glycosylated protein component of the ECM, was downregulated in iPSC-ECs from d-TGA patients. Interestingly, *Hspg2*-null embryos presented with d-TGA [11], supporting the important role of this gene and genes related to ECM in the development of the disease.

Genetic polymorphisms in *GJA4* have been previously related to coronary artery disease [28]. This gene was downregulated in iPSC-ECs from d-TGA patients, suggesting that both *Tbx2* and *GJA4* might be implicated in the development of d-TGA.

Regarding the downregulated genes expressed in EC from patients with d-TGA, we found several different genes involved in Notch signaling. One of these genes, *JAG2*, is directly implicated in angiogenesis [29], and *DLL4* is required for normal arterial patterning [30]. In addition, *DLL4* has an antiangiogenic effect when it competes with *JAG1* [31]. Thus, different genes involved in Notch signaling could be directly involved in disease development. *CDH5* was also found to be downregulated in iPSC-ECs from patients with d-TGA. CDH5 is a transmembrane cadherin protein required for vascular morphogenesis, as its deficiency impairs vasculogenesis in embryonic stem cells derived from embryoid bodies [32] and in mutant mice embryos [33]. These defects were observed, even though the endothelial cells were well differentiated [33]. The downregulation of *KDR*, which has an important role in endothelial specification and maintenance, was also observed. The activation of this receptor is critical for enhancing the proliferation and survival potential of iPSC-ECs [15].

At the functional level, we observed that endothelial cells derived from patients with d-TGA were unable to form capillary-like tubular networks. ECs are directly implicated in angiogenesis, which plays a crucial role during embryonic development and organogenesis [34]. These findings might be related to the low expression levels of genes involved in the Notch signaling pathway observed in the bioinformatics analysis.

The major limitation of our study is the lack of a multicellular system to simulate d-TGA in vitro. However, we have generated an in vitro model of d-TGA using iPSCs as a disease modeling approach. The gene expression differences between the ECs from control and d-TGA patients might be related to the etiology of this cardiac disease. The results presented in this work contribute to a better understanding of the mechanisms promoting the onset of d-TGA during embryonic development. 

Overall, our results show that ECs from patients with d-TGA display genetic differences versus control counterparts. These changes affect biological processes related to the circulatory system and to vascular development. Accordingly, ECs derived from d-TGA iPSCs could be a good cellular model for the study of this pathology. This work supports the use of patient-specific iPSC-ECs in modeling d-TGA.

## 4. Materials and Methods

### 4.1. Human-Induced Pluripotent Cell Generation

Foreskin fibroblasts from a healthy individual and a patient with d-TGA (Ctrl1 and TGA1) were obtained through ATCC and the Coriell Institute, respectively. Fibroblasts were reprogramed following the method described by Yu in 2007 [35], using a lentiviral vector expressing the *Oct4, Nanog, Lin 28 and Sox2* genes. Different clones were obtained and were validated using the following assays: alkaline phosphatase, surface marker expression, pluripotency gene expression, transgene silencing, proviral integration and in vitro and in vivo differentiation (data not shown).

Ctrl2 line (IMR90-4) was purchased from WiCell Research Institute (Madison, WI, USA) and was reprogramed using viral transfection (*Oct4, Sox2, Nanog and Lin28*). The iPSC-TGA line TGA2 (SCVI-235) was generated from peripheral blood mononuclear cells [36] and was obtained from the Stanford Cardiovascular Institute iPSC Biobank (Stanford, CA, USA). 

### 4.2. Cell Culture

All pluripotent cell lines and differentiation cultures were maintained in Matrigel (Corning, Bedford, MA, USA) pre-coated plates at 37 °C in a Thermo Forma 370 Steri Cycle Incubator (Thermo Fisher Scientific, Walthman, MA, USA) with 5% CO_2_ and 21% O_2_ (21% O_2_). Cells were grown with mTeSR1 medium (Stemcell Technologies, Vancouver, Canada) and were passaged at a 1:10 ratio when ~80% confluency was reached (after about 4 days) using Accutase solution (Stemcell Technologies).

### 4.3. Endothelial Differentiation

Endothelial cell differentiation was induced following the protocol described by Prasain et al. in 2014 [15]. Briefly, cells were seeded at a density of 6250 cm^−2^ onto a Matrigel pre-coated 24-well plate (day-2) and were incubated for 2 days in mTeSR1 medium. On day 0, cells were cultured with Stemline^®^ II Hematopoietic Stem Cell Expansion Medium (Sigma-Aldrich, Darmstadt, Germany) containing 10 ng/mL of Activin A (R&D Systems, Minneapolis, MN, USA), FGF-2 (Tebu-bio, Yvelines, France), BMP-4 (R&D Systems, Minneapolis, MN, USA) and VEGF (Peprotech, London, UK) for 1 day to initiate the differentiation. The following day, the medium was replaced and cells were incubated with the same concentration of FGF-2, BMP-4 and VEGF (differentiation medium) for 11 days. On day 12, FACS was performed to isolate endothelial cells.

### 4.4. Fluorescence-Activated Cell Sorting

Cells were detached by trypsinization and filtered through a 40-µm filter to form a single cell suspension. Cells were resuspended in 130 µL of PBS, supplemented with 1% fetal bovine serum and were incubated with anti-human platelet/endothelial cell adhesion molecule-1 (*CD31*; *PECAM-1*, Alexa Fluor 488-conjugated, BD-Pharmingen, San Diego, CA, USA) and Neuropilin-1 (*NRP-1*, APC-conjugated, Miltenyi Biotech, Bergisch Gladbach, Germany). Double-positive CD31+/NRP-1+ cells were isolated by FACS and were expanded in collagen-coated T25 flasks until confluence [37] in Endothelial Cell Growth Medium-2 (EC medium) BulletKit^TM^ (Lonza, Basel, Switzerland). Experiments were performed from day 19 of differentiation and cells were maintained in culture for no more than 5 passages.

### 4.5. Tube Formation Assay 

iPSC-ECs were plated onto Matrigel-coated wells to measure tube formation, as described [38]. Briefly, 20 × 10^4^ cells were seeded per well into 96-well plates and were incubated for 3 h. Images were acquired using an inverted microscope (Leica DM6000, Leica Microsystems, Wetzlar, Germany) with a 10× magnification and were analyzed with WimTube online software (WimTube: Tube Formation Assay Image Analysis. Release 4.0. https://www.wimasis.com/en/WimTube, accessed on 06 December 2021 ). Five images from independent experiments were analyzed for each iPSC-EC line. 

### 4.6. RNA Extraction and Quantitative Real-Time PCR

RNA was isolated from samples on different days of EC differentiation (days 0, 5, 12 and 19) using the RNeasy Plus kit (Qiagen, Dusseldorf, Germany). cDNA was produced using a High-Capacity RNA-to-cDNA Kit (Applied Biosystems, Walthman, MA, USA), and real-time PCR (qPCR) was performed using the LightCycler 480 SYBR Green I Master Kit (Roche Life Science, Switzerland) on a ViiA 7 Real-Time PCR System (Applied Biosystems, Walthman, MA, USA). Primers were provided by Condalab (Madrid, Spain) and glyceraldehyde-3-phosphate dehydrogenase (*GAPDH*) was used as housekeeping control. Reactions were performed in triplicate.

### 4.7. RNA-Seq

Libraries were prepared using the TruSeq Stranded Total RNA Library Prep Kit with Ribo-Zero Human/Mouse/Rat Kit (Illumina, San Diego, CA, USA). Briefly, 300 ng of total RNA was used for ribosomal RNA depletion. Then, ribosomal-depleted RNA was fragmented for 4.5 min at 94 °C. The remaining steps of the library preparation were followed according to the manufacturer’s instructions.

Libraries were analyzed on an Agilent Technologies 2100 Bioanalyzer system, using the Agilent DNA 1000 chip to estimate the quantity and validate the size distribution, and were then quantified by qPCR using the KAPA Library Quantification Kit KK4835 (Roche Life Science, Switzerland). 

FASTQ files were treated with Cutadapt and aligned with Kallisto to obtain the count table for each gene. Differential expression analysis of RNA-seq expression profiles was performed with the EdgeR package [39]. GO enrichment analyses were performed using g:Profiler [40,41].

### 4.8. Interactome Analysis

Analysis was performed as described [21]. Complete interactome was identified, and GSEA was performed to detect significant GO biological processes, molecular functions and cellular components [40].

### 4.9. Statistical Analysis

Results are represented as the mean ± standard deviation (SD). Comparisons between Ctrl and TGA groups were performed with the non-parametric Mann–Whitney U test, one-way ANOVA and necessary post hoc analysis. Analyses were conducted with GraphPad Prism 8^®^ software (GraphPad Software Inc., La Jolla, CA, USA). Differences were considered statistically significant at *p* < 0.05, with a 95% confidence interval.

## Figures and Tables

**Figure 1 ijms-22-13270-f001:**
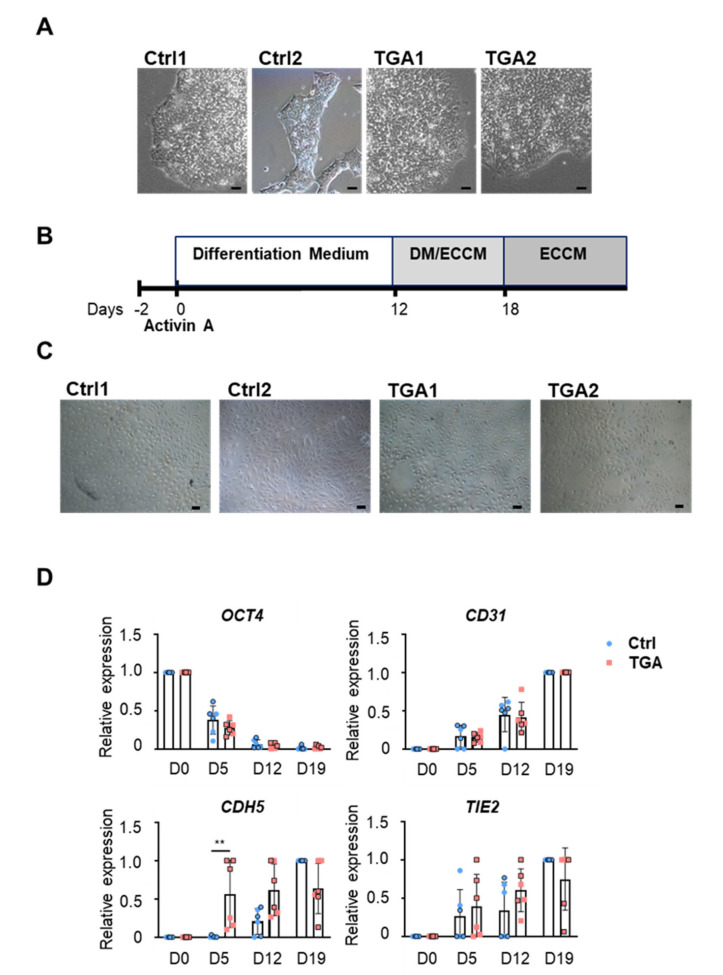
Differentiation of iPSC lines from control and d-TGA patients. (**A**) Representative brightfield images of undifferentiated iPSC colonies on Matrigel. (**B**) Protocol used for differentiation of iPSC lines to EC using differentiation medium (DM) and endothelial cell culture medium (ECCM). (**C**) Representative brightfield images of iPSC-ECs at day 19. Images were recorded using a 10× objective, scale bar = 10 µm. (**D**) Expression of genes involved in EC differentiation measured by qPCR in control (blue, ● Ctrl1, ○ Ctrl2) and d-TGA lines (red, ■ TGA1, □ TGA2). Samples were obtained at days 0, 5, 12 and 19 of EC differentiation. Data were normalized to *GAPDH* expression and are represented as mean ± SD. Three independent differentiations were performed for each iPSC line. ** *p* < 0.01, by Mann–Whitney U test. Scale bar = 10 µm.

**Figure 2 ijms-22-13270-f002:**
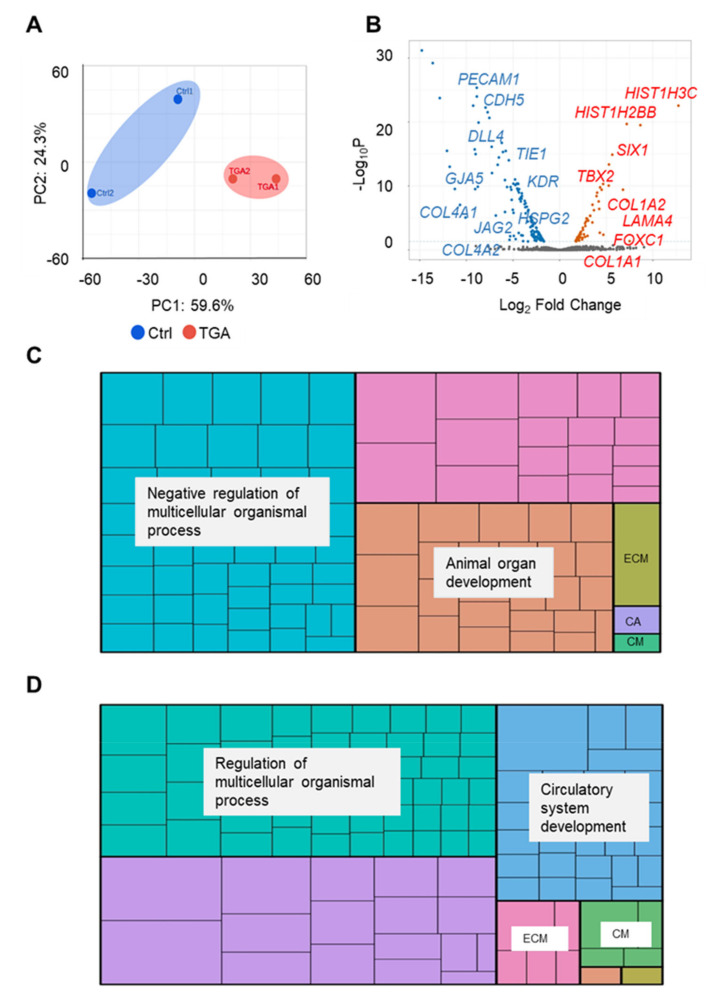
Transcriptome analysis of iPSC-ECs. (**A**) Principal component analysis of component 1 *versus* component 3. (**B**) Volcano plot representing the distribution of log2 gene ratio between iPSC from d-TGA patients *versus* controls *versus* log10 gene intensity. Down- and upregulated genes are plotted in blue and red, respectively. (**C**) Treemap diagram of biological processes of upregulated genes overrepresented in iPSC-ECs from d-TGA *versus* control lines using REVIGO after g:Profiler gene analysis. (**D**) Treemap diagram of biological processes of downregulated genes, overrepresented in iPSC-ECs from d-TGA *versus* control lines using REVIGO after g:Profiler gene analysis. ECM: extracellular matrix organization, CM: cell motility and CA: cell adhesion. GO processes in non-labelled superclusters correspond to non-classified biological processes.

**Figure 3 ijms-22-13270-f003:**
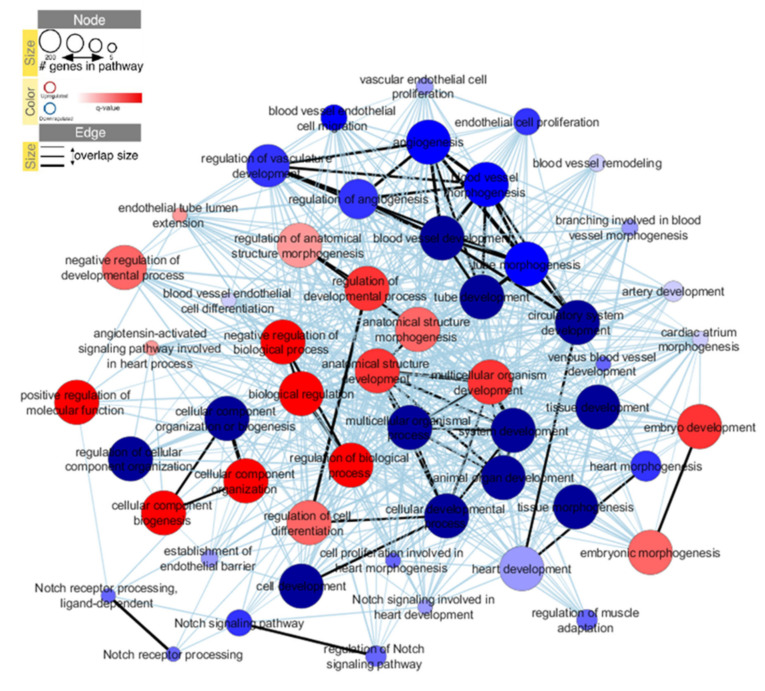
Graphical representation of the iPSC-EC interaction network based on RNA-seq data sets. Biological processes are represented as dots, where the size depends on the number of genes identified in each biological process. Upregulated and downregulated biological processes are represented in red and blue, respectively. The size of the nodes is represented as a function of the number of genes involved in each of the pathways and the colors as a function of the *p*-value. Stronger associations are now represented with black lines, while the others are colored in light blue.

**Figure 4 ijms-22-13270-f004:**
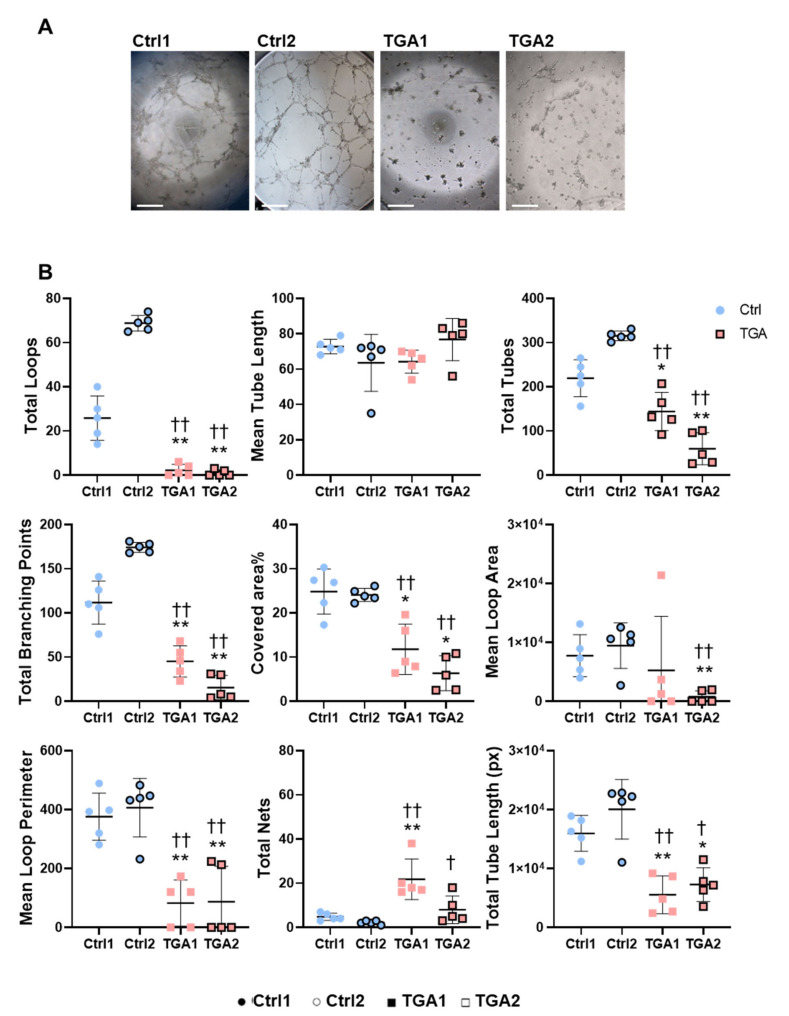
Functional analysis of iPSC-ECs. (**A**) Representative images of tube formation assay 3 h after seeding iPSC-ECs from control and d-TGA lines. (**B**) Quantification of total loops, total mean tube length, total tubes, total branching points, covered area (%), mean loop area and mean loop perimeter, total nets and mean tube length from images in control (blue, ● Ctrl1, ○ Ctrl2) and d-TGA (red, ■ TGA1, □ TGA2) lines. Data are represented as mean ± SD, results are expressed in arbitrary units (*n* = 5, * *p* < 0.05, ** *p* < 0.01 by Mann–Whitney U test in Ctrl1 *versus* TGA1 or TGA2 and † *p* < 0.05, †† *p* < 0.01 in Ctrl2 *versus* TGA1 or TGA2).

**Table 1 ijms-22-13270-t001:** Gene ontology biological processes associated with differentially upregulated genes in iPSC-ECs from patients with transposition of the great arteries *versus* healthy controls.

GO_ID	Term	*p*-Value
GO:0009887	Animal organ morphogenesis	1.59 × 10^−12^
GO:0048513	Animal organ development	1.84 × 10^−10^
GO:0043062	Extracellular structure organization	5.34 × 10^−9^
GO:0048856	Anatomical structure development	1.95 × 10^−6^
GO:0051093	Negative regulation of developmental process	3.13 × 10^−6^
GO:0009790	Embryonic development	9.05 × 10^−5^
GO:0045995	Regulation of embryonic development	1.10 × 10^−3^
GO:0072359	Circulatory system development	4.52 × 10^−3^
GO:0009792	Embryonic development ending in birth or egg hatching	6.98 × 10^−3^
GO:0060560	Developmental growth involved in morphogenesis	2.60 × 10^−2^
GO:0048568	Embryonic organ development	2.60 × 10^−2^
GO:0035148	Tube formation	3.10 × 10^−2^
GO:0048562	Embryonic organ morphogenesis	4.81 × 10^−2^

**Table 2 ijms-22-13270-t002:** Gene ontology biological processes associated with differentially downregulated genes in iPSC-ECs from patients with transposition of the great arteries *versus* healthy controls.

GO_ID	Term	*p*-Value
GO:0072359	Circulatory system development	5.44 × 10^−15^
GO:0007275	Multicellular organism development	3.94 × 10^−12^
GO:0001568	Blood vessel development	6.28 × 10^−12^
GO:0048646	Anatomical structure formation involved in morphogenesis	7.67 × 10^−12^
GO:0048514	Blood vessel morphogenesis	7.67 × 10^−12^
GO:0072358	Cardiovascular system development	7.67 × 10^−12^
GO:0048856	Anatomical structure development	2.86 × 10^−10^
GO:0035239	Tube morphogenesis	3.65 × 10^−10^
GO:0032502	Developmental process	4.03 × 10^−10^
GO:0035295	Tube development	1.07 × 10^−9^
GO:2000181	Negative regulation of blood vessel morphogenesis	1.09 × 10^−5^
GO:0022603	Regulation of anatomical structure morphogenesis	1.10 × 10^−5^
GO:0007507	Heart development	1.42 × 10^−5^
GO:0050793	Regulation of developmental process	1.37 × 10^−4^
GO:0051093	Negative regulation of developmental process	1.10 × 10^−3^
GO:0003230	Cardiac atrium development	5.46 × 10^−3^
GO:1903587	Regulation of blood vessel endothelial cell proliferation involved in sprouting angiogenesis	1.48 × 10^−2^

**Table 3 ijms-22-13270-t003:** Gene ontology biological processes identified from the interactome analysis of upregulated genes in iPSC-ECs from patients with transposition of the great arteries *versus* healthy controls.

GO_ID	Term	*p*-Value
GO:0050789	Regulation of biological process	1.39 × 10^−11^
GO:0050793	Regulation of developmental process	7.00 × 10^−9^
GO:0007275	Multicellular organism development	3.40 × 10^−8^
GO:0048856	Anatomical structure development	8.06 × 10^−8^
GO:0048513	Animal organ development	3.38 × 10^−7^
GO:0009790	Embryonic development	8.96 × 10^−7^
GO:0009653	Anatomical structure morphogenesis	6.32 × 10^−5^
GO:0048598	Embryonic morphogenesis	4.82 × 10^−3^
GO:0051093	Negative regulation of developmental process	1.50 × 10^−2^
GO:0086098	Angiotensin-activated signaling pathway involved in heart process	3.62 × 10^−2^

**Table 4 ijms-22-13270-t004:** Gene ontology biological processes identified from the interactome analysis of downregulated genes in iPSC-ECs from patients with transposition of the great arteries *versus* healthy controls.

GO_ID	Term	*p*-Value
GO:0051128	Regulation of cellular component organization	1.91 × 10^−31^
GO:0048869	Cellular developmental process	6.91 × 10^−23^
GO:0048731	System development	3.88 × 10^−22^
GO:0009888	Tissue development	1.99 × 10^−19^
GO:0048513	Animal organ development	4.43 × 10^−15^
GO:0032501	Multicellular organismal process	3.89 × 10^−13^
GO:0048729	Tissue morphogenesis	5.28 × 10^−12^
GO:0072359	Circulatory system development	8.95 × 10^−9^
GO:0001568	Blood vessel development	3.60 × 10^−8^
GO:0035295	Tube development	9.09 × 10^−8^
GO:0048514	Blood vessel morphogenesis	9.56 × 10^−8^
GO:0001525	Angiogenesis	1.35 × 10^−7^
GO:0035239	Tube morphogenesis	2.12 × 10^−7^
GO:0007219	Notch signaling pathway	7.34 × 10^−6^
GO:0045765	Regulation of angiogenesis	9.75 × 10^−6^
GO:1901342	Regulation of vasculature development	1.80 × 10^−5^
GO:0009790	Embryonic development	6.51 × 10^−5^
GO:0003007	Heart morphogenesis	7.34 × 10^−4^
GO:0007220	Notch receptor processing	2.93 × 10^−3^
GO:0035333	Notch receptor processing, ligand-dependent	5.73 × 10^−3^
GO:0060841	Venous blood vessel development	5.95 × 10^−3^
GO:0061028	Notch signaling involved in heart development	1.06 × 10^−2^
GO:0060837	Blood vessel endothelial cell differentiation	2.16 × 10^−2^
GO:0060840	Artery development	2.88 × 10^−2^
GO:0001974	Blood vessel remodeling	3.36 × 10^−2^

## Data Availability

The data presented in the study are deposited in the GEO database repository. The accession number is GEO:GSE186803.

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
