# Peer review of "Modeling Transposition of the Great Arteries with Patient-Specific Induced Pluripotent Stem Cells"

_ijms, 2021, doi:10.3390/ijms222413270_

Round 1

Reviewer 1 Report

The use of patient-specific iPSC-derived cells is significant and provides an initial disease model that may be expanded with the use of 3D co-cultures. A strength of the approach is evaluating the gene profiling results in a functional assay for tubular formation. The manuscript could be further improved with better representation of figures 2 C and D and figure 3. 

in Figure 2 C and D - I am confused by the representation and the significance. Please provide better definition in the figure and text.

  1. Do the white borders indicate different categories? if so does the lower right grouping part of a different category or the same category as animal organ development?
  2. the color bar needs to be better defined. Does bold red indicate most highly over- and under-expressed in C and D, respectively and there is nothing in bold blue indicating what? 
  3. Are there X- and Y- axis to indicate how the genes are clustered

Figure 3 is difficult to decipher.

  1. the color variations are hard to see. Using different colors to indicate p-value ranges may be easier to understand. 
  2. What is the overall significance as it appears many circles are the same size and color variation? Indicate stronger links with darker line or the like. 

Table 3 and 4 - how are the GO pathways different (or not different) for the interactome analysis versus the normal analysis and how or why is this significant for the same data set?   

Author Response

Dear reviewer,

Please see attached the responses to your comments. We hope this help to better understand the manuscript. Please, don't hesitate to contact us in case you need further clarifications.

Warm regards,

Reviewer 1

The use of patient-specific iPSC-derived cells is significant and provides an initial disease model that may be expanded with the use of 3D co-cultures. A strength of the approach is evaluating the gene profiling results in a functional assay for tubular formation. The manuscript could be further improved with better representation of figures 2 C and D and figure 3. 

in Figure 2 C and D - I am confused by the representation and the significance. Please provide better definition in the figure and text.

  1. Do the white borders indicate different categories? if so does the lower right grouping part of a different category or the same category as animal organ development?
  2. the color bar needs to be better defined. Does bold red indicate most highly over- and under-expressed in C and D, respectively and there is nothing in bold blue indicating what? 
  3. Are there X- and Y- axis to indicate how the genes are clustered

Thank you for the comment; we agree with the reviewer that the representation of Treemaps in figures 2 C and D were confusing. Figures C and D have been simplified to make them easier to understand. To make the Treemaps, we only take into account individual GO with the corresponding p-value. Colors have been changed, and now each supercluster has a unique color, and the size of the boxes depends on the p-value of each GO.

In the new figures, superclusters names are indicated. Colored boxes without title correspond to GOs classified as “null” that groups GOs with no particular relevance for membership of any supercluster.

Figure 3 is difficult to decipher.

We have made some modifications to Figure 3. We hope this help to see the differences better. 

  1. the color variations are hard to see. Using different colors to indicate p-value ranges may be easier to understand. 

We agree with the reviewer that color variations were difficult to visualize. We have increased the contrast between nodes to make it easier to understand.

  1. What is the overall significance as it appears many circles are the same size and color variation? Indicate stronger links with darker lines or the like. 

Nodes are represented in the function of the enrichment values. Stronger associations are now represented with black lines while the others are coloured in light blue.

Table 3 and 4 - how are the GO pathways different (or not different) for the interactome analysis versus the normal analysis and how or why is this significant for the same data set? 

With the analysis of the interactome we are analyzing not only the DEG genes but also the portion of the interactome affected by the disease. This analysis includes both the genes that appear in DEGs and therefore have a different expression in diseased/healthy individuals and the intermediate genes that are involved in the same processes (as they are linked in the interactome).  This analysis allows us to investigate in a more detailed and less biased way the processes involved since the DEGs selected has a p-value <0.05.

Reviewer 2 Report

This is an interesting study on modeling the great arteries using patient specific iPSCs. The authors analyzed the transcriptome profiling and bioinformatics in endothelial cells, and revealed some significant downregulation of biological processes including Notch signaling. There are some suggestions before publication.

  1. After sorting, the isolated ECs may need to be confirmed by immunostaining.
  2. The changes of important genes that were mentioned in section 2.2 need to be confirmed by qualitative PCR.
  3. Figure 4B, as you have 2 individuals, please split the control and TGA to 2, respectively and then compare. 
    Was the mean tube length changed in TGA? It looks there was a significant change in the representative picture of Fig. A. 
  4. Figure 1 needs scale bars.

Author Response

Dear reviewer,

Please find attached the responses to your comments. We hope this help to better understand the manuscript. Please, don't hesitate to contact us in case you need further clarifications.

Warm regards,

Imelda Ontoria-Oviedo
